# Patterns of Perceived Indoor Environment in Danish Homes

**DOI:** 10.3390/ijerph191811498

**Published:** 2022-09-13

**Authors:** Stine Kloster, Anne Marie Kirkegaard, Michael Davidsen, Anne Illemann Christensen, Niss Skov Nielsen, Lars Gunnarsen, Annette Kjær Ersbøll

**Affiliations:** 1National Institute of Public Health, University of Southern Denmark, Studiestraede 6, 1455 Copenhagen K, Denmark; 2Department of the Built Environment, Aalborg University, A.C. Meyers Vaenge 15, 2450 Copenhagen SV, Denmark

**Keywords:** perceived indoor environment, annoyances, housing condition, environmental epidemiology, clustering, factor analysis, latent class analysis

## Abstract

The indoor environment is composed of several exposures existing simultaneously. Therefore, it might be useful to combine exposures into common combined measures when used to assess the association with health. The aim of our study was to identify patterns of the perceived indoor environment. Data from the Danish Health and Morbidity Survey in the year 2000 were used. The perceived indoor environment was assessed using a questionnaire (e.g., annoyances from noise, draught, and stuffy air; 13 items in total). Factor analysis was used to explore the structure of relationships between these 13 items. Furthermore, groups of individuals with similar perceived indoor environment were identified using latent class analysis. A total of 16,688 individuals ≥16 years participated. Their median age was 46 years. Four factors were extracted from the factor analysis. The factors were characterized by: (1) a mixture of items, (2) temperature, (3) traffic, and (4) neighbor noise. Moreover, three groups of individuals sharing the same perception of their indoor environment were identified. They were characterized by: a low (n = 14,829), moderate (n = 980), and large number of annoyances (n = 879). Observational studies need to take this correlation and clustering of perceived annoyances into account when studying associations between the indoor environment and health.

## 1. Introduction

The indoor environment in homes is important for health and well-being [1,2,3] since we spend around two thirds of our time indoors at home [4,5]. However, the influence of the indoor environment on health has not been investigated to the same extent as the outdoor environment [6,7]. 

The indoor environment is often defined as components related to thermal, visual, and acoustic comfort as well as indoor air quality [8,9]. Risk factors associated with an impaired indoor environment are, among others, thermal discomfort [8], limited daylight [9], noise [10], chemicals [1], allergens [1], environmental tobacco smoke from primary [1] and second-hand smoke [1,2], and presence of dampness and mold [1,11,12]. These factors, among others, are causes of the sick building syndrome [13]. Annoyances from poor indoor air quality might be the first indicator of a problem with the indoor environment. Subjectively evaluated indoor air quality has shown to be a good indicator of objectively assessed indoor environments in Swedish homes [14]. 

The indoor environment is a complex quantity, composed of several different coexisting exposures. Therefore, people are exposed to multiple indoor environmental exposures simultaneously. Hence, in single-exposure studies, it is often unclear if an observed association reflects the effect of the single exposure examined, or if it acts as a surrogate for other exposures originating from the same source and thereby correlates with the single exposure examined [15]. Moreover, single-exposure studies cannot illuminate the mixture and interaction of different exposures [15]. Instead, there might be population subgroups sharing similar exposure profiles, a specific pattern in the indoor environment exposure, or an accumulation of different exposures that might be of importance for health and well-being [15,16]. Recent research has demonstrated that type of housing, housing age, ownership, and source of particle pollution correlate; e.g., living in apartments was correlated with exposure to cooking fumes, and newer buildings were less likely to have mold, etc. [17]. These patterns and correlations of exposures are essential to understand to study the association with human health properly. Therefore, it might be useful to combine indoor exposures into different measures of the indoor environment when used to assess the association with well-being and health. Hence, statistical approaches that can handle multiple correlated exposures might be a way forward when studying the indoor environment in relation to human health. Thus, the aim of our study was to characterize the perceived indoor environment in Danish homes. Moreover, the aim was to identify potential patterns of a perceived indoor environment.

Large cohort studies on the association between indoor environment and health outcomes are lacking [1]. However, before studying the association between indoor exposures and long-term health in large cohorts, the patterns and correlations of the indoor exposures are important to understand. 

## 2. Materials and Methods

### 2.1. Study Design and Setting

Cross-sectional data from the Danish Health and Morbidity Survey in the year 2000 were used. The sampling procedure and data collection have been described in detail elsewhere [18,19]. In brief, a nationally representative random sample of 22,486 adults (age ≥ 16 years) was drawn from the Danish Civil Registration System. The sample was stratified to include at least 1000 individuals from each of the 16 administrative units which existed at the time. A total of 16,688 individuals participated, corresponding to a response rate of 74.2%. Participating individuals were representative of the Danish population in the year of inclusion [20]. Data were collected in three waves during February, May, and September in 2000 by trained interviewers in the home of the respondent [18]. 

### 2.2. Assessment of Demographic Data and Health Behavior

Information about sex, cohabitation status (married/cohabitating or living alone), and age categories (16–24, 25–44, 45–66, 67–79, and ≥80 years) was obtained from the Civil Registration System [21]. Educational information was retrieved from the Danish Education Register [22] at Statistic Denmark and classified according to the International Standard Classification of Education System (ISCED 2011) [23]. Based on highest attained education, educational level was categorized into three groups: elementary (preprimary, primary, and lower secondary; ISCED level 1–2), short (upper secondary and postsecondary; ISCED level 3–4), and medium/long (tertiary education; ISCED level 5–8). If information about educational level was not available in the register, information from the questionnaire was used whenever available.

Information about body mass index (BMI), smoking status, exposure to second-hand smoking, and self-reported health was obtained from the questionnaire. BMI was calculated based on self-reported weight and height. Smoking status was grouped based on two questions: (1) Do you smoke? and (2) Have you smoked previously? Individuals were categorized as a current smoker if they answered “yes, I smoke daily” or “yes, but not every day” to the first question (“Do you smoke”?). Former smokers were individuals who stated that they did not smoke and reported that they have been smoking previously. Never-smokers were individuals who answered no to both questions. Exposure to second-hand smoking was grouped into four groups based on the number of hours per day spent in the residence while people smoked (0, 1, 2, and ≥3 h). Self-rated health was reported as very good, good, reasonable, poor, and very poor. 

### 2.3. Assessment of Residential Data 

Information about type of dwelling, number of adults (age ≥ 16 years), and number of children (age ≤ 15 years) was obtained from the questionnaire. Type of dwelling was categorized as a detached house, semi-detached and terrace house (2–4 family houses and terrace houses), apartments, farms, and others. 

Information on number of individuals in the family, location of dwelling (city or rural district), size of dwelling in square meters in six categories, and construction period in five categories was retrieved by linkage to Statistic Denmark. Resident density was based on information of dwelling size in square meters and number of individuals in the family from Statistic Denmark. 

### 2.4. Assessment of Perceived Indoor Environment

Perceived indoor environment was based on 13 items: perceived annoyances within the past 14 days (12 items) and placement of dwelling next to a road with through traffic (no/yes) (1 item). 

Perceived annoyances within the past 14 days were assessed by asking individuals whether they had been annoyed by:(1)Too low/high temperatures;(2)Draught;(3)Draught along the floor;(4)Stuffy air;(5)Shock from static electricity;(6)Traffic noise;(7)Noise from installations;(8)Noise from neighbors;(9)Noise from nearby industry;(10)Infrasound or low-frequency sound;(11)Vibration in building (e.g., from traffic);(12)Too little light.

All 12 items had three response options (i.e., “no, not annoyed”, “yes, slightly annoyed”, or “yes, very annoyed”). Since the proportion of individuals answering “yes, slightly annoyed” and “yes, very annoyed” was very low, these groups were collapsed into one group, “yes annoyed”. Lastly, the interviewer registered whether the residence was placed next to a road with through traffic (no/yes). See Appendix A for more details.

### 2.5. Statistical Method

For the descriptive analyses, median and interquartile range (IQR) were used for continuous variables and counts with proportions were used for categorical variables. 

#### 2.5.1. Factor Analysis

Factor analysis was used to explore the structure of relationships between the 13 items measuring self-reported indoor environment. To assess the adequacy of the sample for factor analysis, we used the Kaiser–Meyer–Olkin measure where values above 0.6 were considered tolerable [24]. Self-reported annoyances were included in three categories as they yielded the highest Kaiser–Meyer–Olkin value as compared to annoyances included as binary variables.

To determine the number of latent factors that should be retrained (for further analysis) we used the eigenvalue, scree plot, and proportion of variance accounted for. In practice, the scree plot of the eigenvalues was evaluated to determine the “break” where the curve was clearly levelling off indicating the number of factors to include. Moreover, items loading on a factor should share a conceptual meaning [24]. To allow correlations between factors, an oblique rotation (promax) was used. Items with factor loading <0.4 were excluded. 

#### 2.5.2. Latent Class Analysis

We defined groups of individuals with similar indoor environment using latent class analysis (LCA) [25]. LCA with one to four classes was fitted to the 13 items about indoor environment. Variables were included in the models as binary (0 = no annoyance/no road traffic or 1 = yes annoyance/road traffic). The final model was selected based on the Bayesian information criteria (BIC) and log likelihood (LL). The model with the lowest BIC and LL value was preferred. A scree plot of the BIC values was used to illustrate BIC for an increasing number of classes. Based on the selected model, the probability of belonging to each group was obtained for each respondent, and the respondent was assigned to the group with the highest probability. 

#### 2.5.3. Summary Score

We created a summary score of perceived indoor environment by summarizing all 13 items. Annoyances ranged from 0 to 2, and placement of dwelling next to a road with through traffic ranged from 0 to 1. Therefore, the summary score could range from 0 (no problems) to 25 (several problems). Furthermore, the summary score was summarized within each class identified in the LCA. 

Moreover, a summary score was constructed based on the items included in each of the retained factors in the factor analysis and named accordingly. In this case, each of the summary scores were based on different numbers of items. To ease comparability between each summary score, each of them was standardized to range from 0 to 1. 

All data management and statistical analyses were conducted using STATA software version 17.0.

## 3. Results

A total of 16,688 individuals participated in the Danish Health and Morbidity Survey in the year 2000. In all, 126 individuals had missing information on one or more of the included items (Figure 1). Moreover, it was impossible to link individuals with information about the residence at Statistic Denmark for 647 individuals. However, this was only of importance in the descriptive analyses and did not influence the included number for individuals in the factor analysis and LCA. 

The characteristics of individuals and their dwellings are described in Table 1. A total of 49.1% of individuals were male and the majority of individuals were married or cohabiting (70.3%). The median age was 46 years with 4.6% of the individuals being 80 years old or more. More than half of the population were in the normal weight range (BMI 18.5–24.9), 39.3% were never smokers, and 77.7% reported good or very good self-rated health. More information about the health status can be found in the study by Kjøller and Rasmussen 2000 [20]. Most individuals lived in a city area (>200 inhabitants), and detached houses were the most common type of dwelling (51.4%).

A total of 26.5% of individuals reported at least one of the 12 annoyances within the past 14 days. The proportion of each perceived annoyance is described in Table 2. Annoyances from too low or high temperature, draught along the floor, and noise from neighbors and traffic were among the most common annoyances reported with 5.8%, 6.9%, 7.4%, and 5.9%, respectively. Furthermore, these annoyances were reported more often among individuals living in apartments than individuals living in detached houses (e.g., 19.3% versus 3.0% for neighbor noise and 11.5% versus 4.4% for draught along the floor), whereas shock from static electricity, infrasound/low-frequency sound were among the least reported (1.0% and 0.7%, respectively). Additionally, 37.7% lived next to a road with through traffic, almost half of the population lived together with individuals who smoked (48.5%), and 40% were exposed to second-hand smoke. 

We retrained a model with four factors after inspection of the scree plot (Figure 2). The curve flattened out after four factors indicating that a four-factor model was appropriate. The item ‘stuffy air’ was excluded due to low factor loadings. The final model included four factors and the factor loadings are presented in Table 3. The first factor had high loadings on shock from static electricity, noise from installations, noise from nearby industry, infrasound/low-frequency sound, and dwelling having too little light; the factor was named ‘*mixed*’. The second factor had high loadings on too low/high temperatures, annoyances by draught, or draught along the floor; the factor was named ‘*temperature*’. Factor 3 was characterized by loading high on items measuring annoyance by traffic and named thereafter (annoyances from traffic noise, vibrations in building, and residence placed to a road with through traffic). The last factor, factor 4, was characterized by annoyances from noise from neighbors only and named *‘neighbor noise’*. 

The LCA did not converge with more than three classes. The results from the LCA showed that the BIC value favored the three-class model over the two-class model (Figure 3). Therefore, the three-class model was selected. The following names were assigned to the three classes: (1) very few annoyances (88.8%, n = 14,829 based on most likely class membership), (2) moderate annoyances (5.9%, n = 980), and (3) many annoyances (5.3%, n = 879). The probabilities for each item within each class are shown in Figure 4.

The total summary score ranged from 0 to 24. The median (IQR) was 1 (0–1) (Table 4). The standardized summary scores calculated based on items included in the four factors, identified in the factor analysis, are presented in Table 4. The median (IQR) summary score in classes 1, 2, and 3 was 0 (0–1), 3 (2–3), and 4 (3–5), respectively (Figure 5).

## 4. Discussion

In this large survey of the indoor environment in Danish homes, we show that different exposures in the perceived indoor environment often are presented simultaneously. 

### 4.1. Factor Analysis

The 13 items, related to indoor environment, assessed in the current study can be summarized in four factors. The correlations between items are as expected and items clustering together have conceptual meaning; e.g., factor 2 (temperature) had high loadings on too low/high temperature, draught, and draught along the floor. As shown in a study by Wang and Norbäck (2021), these items might all be related to thermal insulation [14]. In line with our results, Wang and Norbäck (2021) identified too low/high temperature and unstable temperature as correlating into one factor [14]. However, they also identified a second factor loading high on perceived stuffy air, dry air, and unpleasant air. In our analysis, stuffy air was not included in a factor, maybe because we did not have any other items that are proxies of the air quality. In a Danish study, characterizing the indoor environment among Danish children, factor analysis was also used [17]. Likewise, they also found several items (behavior, housing characteristics, etc.) to cluster, supporting the importance of knowledge of correlations between exposures when examining the association between the indoor environment and health and well-being. However, the included information differed and results from the identified factors are, therefore, not comparable.

The identified factors can guide us whenever examining the association between indoor environment and risk of diseases. Items separated into different factors will not coexist by default. As an example, noise from neighbors and noise from traffic were separated into distinct factors, indicating that a high reporting of annoyances from neighbor noise does not come together with a high reporting of noise from traffic. This is important information when investigating either of the exposures as risk factors for disease. However, this does not rule out that they can coexist in some cases. Nevertheless, it is possible to examine the two items as separate exposures. Likewise, items which are not included in any of the factors, e.g., stuffy air, can be analyzed as separate exposures without concerns about correlations with other items examined in the current study. 

### 4.2. Latent Class Analysis

Three latent classes were identified by the LCA and named according to the accumulation of annoyances in each group. The first class is characterized by having a low reporting of annoyances in general. This group constitutes the majority of participants. The second group is primarily characterized by annoyances from traffic noise, noise from neighbors, and vibrations in buildings. The last group is characterized by more reported annoyances in general, but especially annoyances from items related to temperature. The three groups indicate a higher accumulation of reported annoyances for each group concurrent with a difference in the dominating items reported in each group. The accumulation of annoyances is supported by the summary score divided into latent classes (Figure 5), which increases steadily for each group. 

Our results from the factor analysis and LCA indicate the importance of considering the correlation and clustering of indoor exposures when studying the association between an indoor environment and health outcomes. 

The identified patterns in the LCA as well as the identified factors in the factor analysis might reflect underlying housing condition, building constructions, building materials, etc., which cause the annoyances reported in the current study. Further research on the patterns of an indoor environment could explore possible risk factors for the observed patterns.

### 4.3. Strength and Limitations

We used a large representative survey sample with extensive information about perceived indoor environment to describe patterns of the perceived indoor environment in Danish homes. Furthermore, comprehensive register-based information about demographic and sociodemographic characteristics and housing conditions at the individual level was included for descriptive purposes. For future research, the cohort can be linked to health outcomes at an individual level. 

The identified factors and latent classes provide important information about the coexistence of exposures and thereby constitute a unique data material to further examine associations between indoor environment and health. Hence, they provide us with important knowledge on when exposures can be examined individually and when they coexist with other exposures. However, for future research and generalizability to other cohorts, more knowledge is needed on whether we describe a general clustering of items, or whether the patterns are specific to our cohort. 

Despite the extensive amount of information about perceived indoor environment included in the current study, other items might also be of importance whenever characterizing the indoor environment, for example, the use of a wood-burning stove, animal related exposures, and use of candle lights, which we do not have information on in the current study. 

We used self-reported information about perceived indoor environments. Therefore, the reported annoyances will probably reflect two things: (1) an indicator of a physical condition in the dwelling or measurable factors; for example, perception of draught, too low temperature, unpleasant odor, dust, and dirt have previously been related to a lower level of thermal insulation [14]; and/or (2) a difference in personal demands of the indoor environment. The threshold of perceived annoyances would probably vary by individual, e.g., women have a lower threshold for detection of odor and thermal discomfort, while older people have a higher threshold [14,26]. Likewise, smokers have been reported to have a higher odor threshold [14,26], and young people report annoyances from noise more often compared to older people [27].

Data collection was conducted in three waves in February, May, and September, which can potentially affect the reported annoyances since some might be affected by season, e.g., the perception of a draught. Therefore, caution should be taken whenever reporting the prevalence of each annoyance. Furthermore, there is a risk of non-response bias despite the high response rate. From later health and morbidity surveys, we know that non-participants often differ from participants in sociodemographic status, gender, and age [28]. This can potentially influence the prevalence of reported annoyances. 

The patterns of exposures described in the current study are based on data collected in the year 2000. Understanding these patterns is important before studying the association between indoor exposures and long-term health. For future research, we will be able to link the cohort described in the current study with data on long-term health in the Danish national health registries (e.g., the Danish National Patient Register [29] and Danish Prescription Registry [30]) at an individual level and thereby study the association between the described patterns of indoor environment and long-term health and well-being with 20 years of follow-up. 

During the past 20 years, some of the examined exposures have probably changed and some have not. Overall, the total number of dwellings in Denmark has increased from about 2.7 million in 2000 [31] to about 3.0 million in 2022 [32]; however, the distribution of housing types is almost similar, i.e., the proportion of multi-story housing accounted for 38.5% in 2000 (without cottages) [31], and 40.0% in 2022 (without cottages) [32]. From later waves of the health and morbidity survey, we know that the prevalence of some annoyances has increased, e.g., noise annoyance from traffic was 5.9% in 2000 and 8.1%, 9.6%, and 9.5% in 2005, 2010, and 2013, respectively [33]. Neighbor noise has changed from 7.4% in 2000 to 15.4% in 2013; however, the change in methodology suggests that the surveys are not comparable [33]. The most substantial change is probably the change in smoking habits, tobacco restriction, etc. The proportion of daily smokers has decreased from 30% in 2000 to 16% in 2017 [34] (p. 15). Likewise, the proportion of people who smoke inside their home has decreased from 18% to 10% in the same period [34] (p. 50). To describe the development in changes in indoor exposure over two decades was beyond the aim of the current study. However, such changes should, like every environmental exposure, be borne in mind whenever studying the association between indoor exposures and health. Nevertheless, we expect the identified patterns to be more independent of time than the single exposure/items included in the factors/latent classes. This, of course, would depend on the mechanism or causes behind the identified patterns. 

## 5. Conclusions

Annoyances from too low or high temperature, draught along the floor, and noise from neighbors and traffic were among the most common annoyances reported. Furthermore, these annoyances were reported more often among people living in apartments than people living in detached houses.

The 13 items of perceived indoor environment could be summarized in four distinct factors or three latent classes. The four factors were characterized by a factor including mixed annoyances, a factor of annoyances related to temperature, a factor of annoyances related to traffic, and one related to noise from neighbors. The three latent classes were characterized by an increased accumulation of annoyances for each class. Overall, most people reported a low number of annoyances (class 1). A smaller group was characterized by annoyances from traffic noise, noise from neighbors, and vibrations in buildings primarily (class 2). Meanwhile, the last group was generally characterized by more reported annoyances with annoyances related to temperature being particularly dominant. 

Epidemiological studies need to take into consideration the high likelihood of coexistence of various exposures related to the indoor environment when examining the association between indoor environment and health. 

## Figures and Tables

**Figure 1 ijerph-19-11498-f001:**
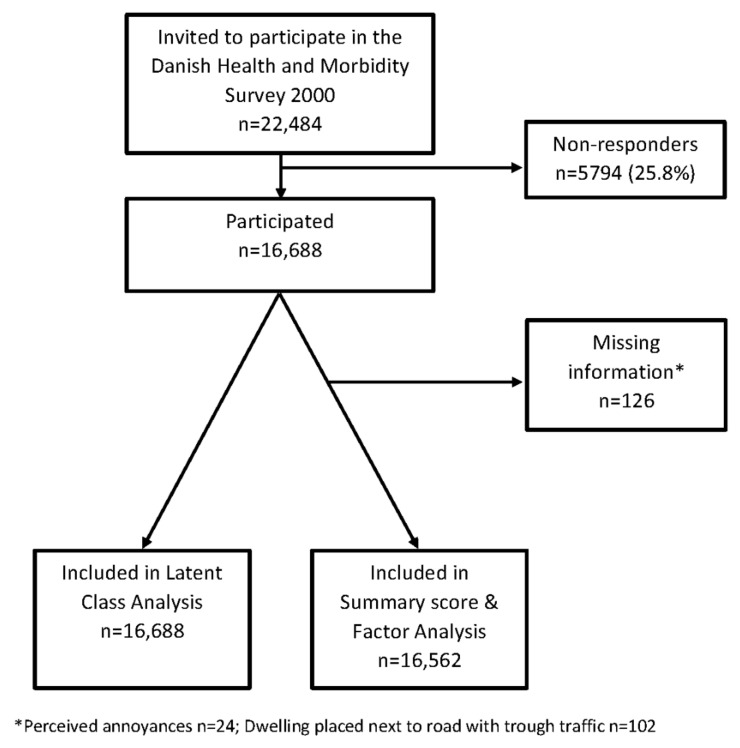
Study flow of participants.

**Figure 2 ijerph-19-11498-f002:**
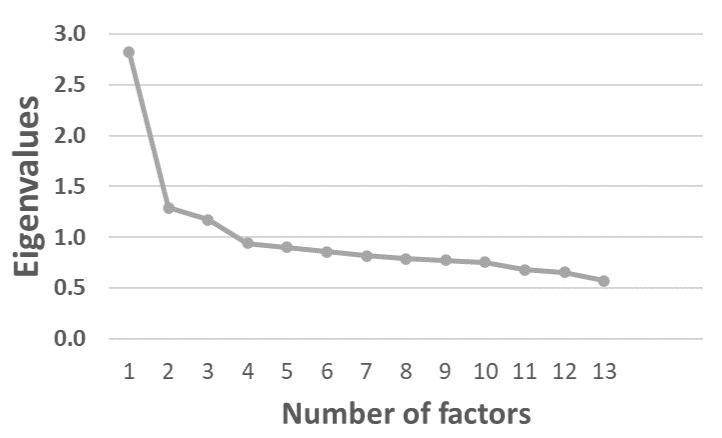
Scree plot of eigenvalues of latent factors extracted from factor analysis. The scree plot shows the eigenvalues of factors from factor analysis, with eigenvalue of the y-axis and the number of factors on the x-axis.

**Figure 3 ijerph-19-11498-f003:**
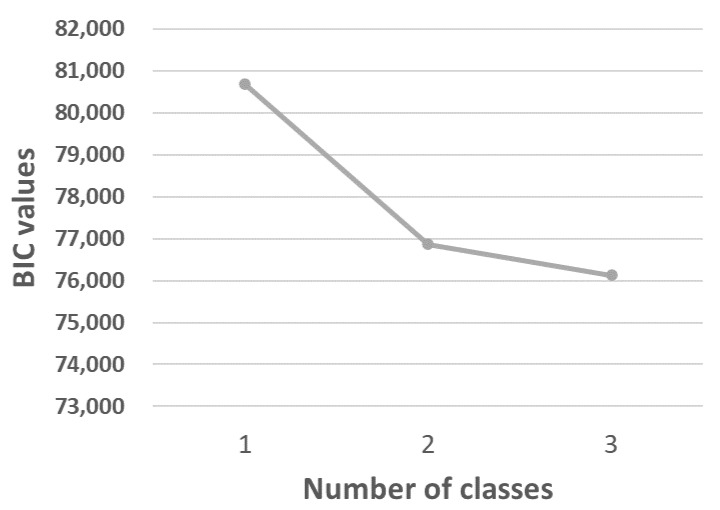
Scree plot of BIC values from Latent Class Analyses with one to three classes.

**Figure 4 ijerph-19-11498-f004:**
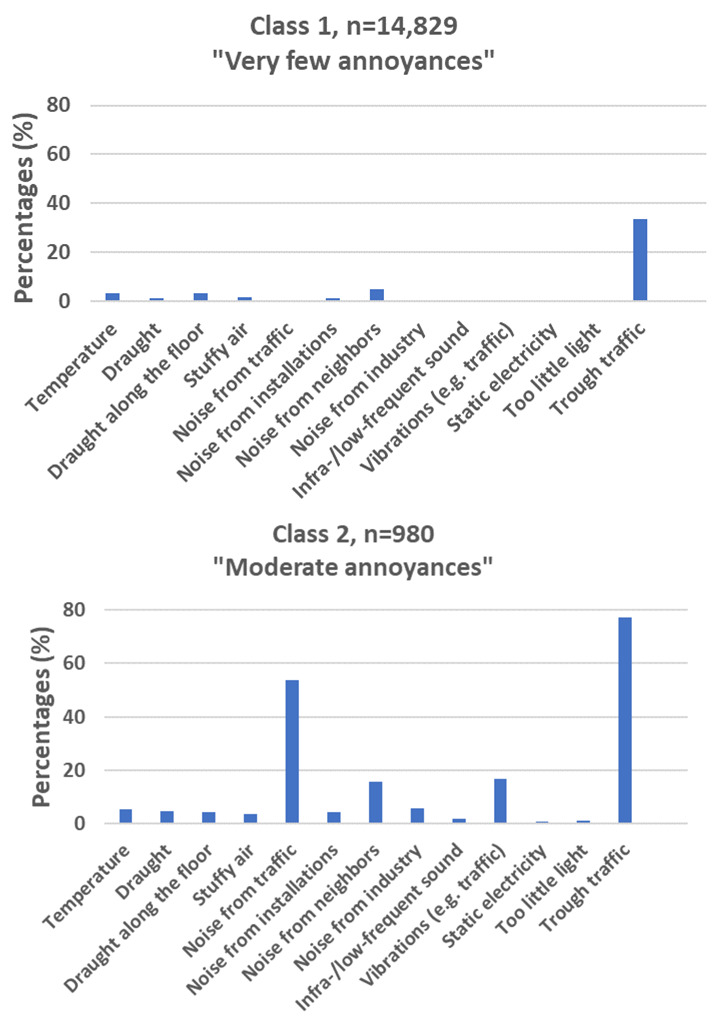
Predicted probabilities for levels of annoyances in homes, and nearness to a road with through traffic from three classes (n = 16,688). Class 1 “Very few annoyances”, Class 2 “Moderate annoyances”, and Class 3 “Many annoyances”.

**Figure 5 ijerph-19-11498-f005:**
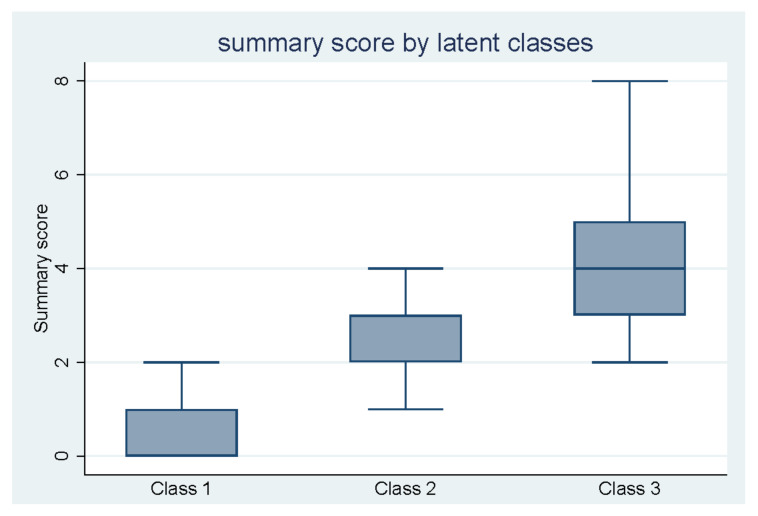
Boxplot comparing the summary score by the three classes identified in the latent class analysis. The blue center line denotes the median value (50th percentile), while the blue box contains the first (25th percentile) to third quartile (75th percentiles) of the dataset. The blue whiskers mark the first and third quartile minus/plus 1.5 times the interquartile range. Values beyond these upper and lower bounds are considered outliers and excluded in the plot.

**Table 1 ijerph-19-11498-t001:** Baseline characteristics of the study population and their dwellings, n = 16,688.

Characteristics	Categories	N	%
Study population			
Sex	Male	8186	49.1
	Missing	0	
Civil status	Married/cohabiting	11,730	70.3
	Living alone	4954	29.7
	Missing	4	0.02
Age (years)	Median (IQR)	16,681	46 (32–59)
	16–24	2187	13.1
	25–44	5817	34.9
	45–66	6042	36.2
	67–79	1876	11.2
	≥80	762	4.6
	Missing	4	0.02
Body mass index (BMI) (kg/m^2^)	<18.5	473	2.9
	18.5–24.9	8962	54.7
	25–29.9	5369	32.7
	≥30	1596	9.7
	Missing	288	1.7
Smoking habits	Current smoker	6188	37.2
	Former smoker	3921	23.5
	Never smoker	6547	39.3
	Missing	32	0.2
Second-hand smoking (hours in residence)	0	9872	59.9
	1	1288	7.8
	2	1078	6.5
	≥3	4254	25.8
	Missing	196	1.2
Educational level	Elementary	6791	40.7
	Short	6726	40.3
	Medium/long	3161	20.0
	Missing	10	0.01
**Dwellings**			
Location of dwelling	City	13,409	83.6
	Land district	2632	16.4
	Missing	647	3.9
Type of dwelling	Detached house	8525	51.4
	Semi-detached house and terrace house	2832	17.1
	Apartment	3428	20.7
	Farm	1363	8.2
	Other	435	2.6
	Missing	105	0.6
Size of dwelling (m^2^)	<50	566	3.5
	50–69	1620	10.2
	70–89	2479	15.5
	90–109	2452	15.4
	110–139	3676	23.0
	≥140	5183	32.3
	Missing	712	4.3
Construction period	<1950	6423	40.2
	1951–1960	1455	9.1
	1961–1972	3498	21.9
	1973–1978	1871	11.7
	≥1978	2738	17.1
	Missing	703	4.2
Number of persons in dwelling	≥16 years (median IQR)		2 (2–2)
	Missing	38	0.2
	<16 years (median IQR)		0 (0–1)
	Missing	<5 *	NA
Resident density (m^2^/person)	Median (IQR)	15,976	52.0 (36.3–70.0)
	Missing	732	4.4

IQR, interquartile range; NA, not applicable * Exact n is not given because of data privacy policy. The exact number is known by the researchers and used in calculations.

**Table 2 ijerph-19-11498-t002:** Frequency of reported annoyances, n = 16,688.

Home Characteristics	Categories	%
**Reported annoyances by**^**a**^:		
Too low/high temperature	Yes	5.8
Draught	Yes	4.3
Draught along the floor	Yes	6.9
Stuffy air	Yes	3.2
Shock from static electricity	Yes	1.0
Traffic noise	Yes	5.9
Noise from installations	Yes	2.7
Noise from neighbors	Yes	7.4
Noise from nearby industry	Yes	1.5
Infrasound or low-frequency sound	Yes	0.7
Vibrations in building (e.g., from traffic)	Yes	2.2
Too little light	Yes	1.5
**Other items**^**b**^:		
Location of dwelling next to a road with through traffic	Yes	37.7

^a^ missing n = 24, ^b^ missing n = 102.

**Table 3 ijerph-19-11498-t003:** Factor analysis of perceived annoyances in homes, and nearness to a road with through traffic, n = 16,349.

	Factors				
Source	1	2	3	4	Uniqueness
	Mixed	Temperature	Traffic	Neighbor Noise	
**Annoyances by:**					
Too low/high temperature		0.61			0.61
Draught		0.77			0.41
Draught along the floor		0.80			0.39
Shock from static electricity	0.73				0.54
Noise from neighbors				0.97	0.15
Traffic noise			0.67		0.42
Noise from installations	0.43				0.71
Noise from nearby industry	0.54				0.64
Infrasound or low-frequency sound	0.78				0.47
Vibrations in building (from, e.g., traffic)			0.51		0.55
Dwelling placed to road with through traffic			0.75		0.41
Too little light	0.45				0.68

**Table 4 ijerph-19-11498-t004:** Summary score of perceived indoor environment. N = 16,562.

		Summary Score		
Factors	N	Mean ± SD	Median	IQR
Overall ^a^	16,562	0.94 ± 1.54	1	0–1
Mixed ^b^	16,562	0.02 ± 0.07	0	0–0
Temperature ^c^	16,562	0.06 ± 0.17	0	0–0
Traffic ^d^	16,562	0.15 ± 0.21	0	0–0.33
Neighbor noise ^e^	16,562	0.07 ± 0.26	0	0–0

^a^ All reported annoyances; ^b^ Factor 1 (Mixed): Shock from static electricity, noise from installations, noise from nearby industry, and infrasound or low-frequency sound; ^c^ Factor 2 (Temperature): Too low/high temperature, draught, and draught along the floor; ^d^ Factor 3 (Traffic): Traffic noise, vibrations in building (e.g., from traffic), and dwelling placed to road with through traffic; ^e^ Factor 4 (Neighbor noise): Noise from neighbors.

## Data Availability

The data that support the findings of this study are available from Statistics Denmark, but restrictions apply to the availability of these data, which were used under license for the current study, and so are not publicly available.

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
