# Peer review of "Patterns of Perceived Indoor Environment in Danish Homes"

_ijerph, 2022, doi:10.3390/ijerph191811498_

Round 1

Reviewer 1 Report (Previous Reviewer 1)

Because the authors provided satsified response to my comments, I suggest the paper shoud be accepted for further publication.

Author Response

We are most grateful for the time you spent on providing suggestions on how we could improve our paper in the first round.

Reviewer 2 Report (New Reviewer)

The authors seek to characterize the perceived indoor environment in Danish homes and identify potential patterns of perceived indoor environments. Overall, the discussion of the manuscript can be further strengthened. I would recommend this manuscript be published after some revisions. My detailed comments are listed below:

1. When the author explains the indoor environment, related to thermal-, visual- and acoustic comfort and indoor air quality, in the ‘Introduction’, they should mention Sick Building Syndrome (SBS). It is a widely accepted term on a global scale. Then IAQ, chemicals, allergies, and molds should be described as SBS causes. Author should not simplify as ‘stuffy air”

2. Regarding IAQ, authors can not associate smoking or secondary smoking with IAQ measurement since smoking is a too strong influential factor and interfere with all the measurements for indoor air pollutants. I recommend authors be clear about this.

3. In ‘Materials and Methods’, I am not 100% sure that Cross-sectional data from the Danish Health and Morbidity Survey in the year 2000 is valid for indoor environment-related building material, etc. 10 years is a long time and many construction methods might be changed and also many global standards might be changed for IAQ, etc. Please justify why the authors used this dataset.

4. Author should stick to IJERPH manuscript format, especially for headings.

3. Section

3.1. Subsection

3.1.1. Subsubsection

Bulleted lists look like this:

·           First bullet;

·           Second bullet;

·           Third bullet.

Numbered lists can be added as follows:

1.        First item;

2.        Second item;

3.        Third item.

5. I found many typos and grammar errors. I hope authors can use professional English editing. Since this paper doesn’t use IJERPH manuscript format with line numbers, I can’t specify the locations. Please correct them accordingly.

6 The quality of all figures can be improved. i.e. vaule on top of each bar, etc. In general, all of them should be more intuitive.

7. The conclusion is not clear to me. When the authors mentioned the identification of the potential patterns of perceived indoor environments, I expected them to explore the interrelationship among indoor environmental factors via statistics. What did the authors find with Latent Class Analysis? Could they be more clear about this?

8. In general, without any new findings with potential patterns of perceived indoor environments, this manuscript used 13 extremely broad definitions of the indoor environment such as 1)   Too low/high temperatures, 2) Draught, 3) Draught along the floor, 4) Stuffy air, 5) Shock from statistic electricity, 6) Traffic noise, 7) Noise from installations, 8) Noise from neighbors, 9) Noise from nearby industry, 10) Infrasound or low-frequent sound, 11) Vibration in building (from e.g., traffic), 12) Too little light, 13) Traffic.

Each topic can be one manuscript. If authors want to publish this manuscript, they should be really focused on the relationship among these factors as they describe in their aims.

Author Response

This manuscript is a resubmission of an earlier submission. The following is a list of the peer review reports and author responses from that submission.

Round 1

Reviewer 1 Report

Based on the analysis on data from the Danish Health and Morbidity Survey in year 2000, perceived indoor environment was assessed by questionnaire and factor analysis was made. Some inteteresting findings were explored. The study is useful to understand perceived indoor environment in Danish. However, because the survey was made in 2000 and the situation in past more than 20 years , the discussion may be changed, the discussion on the applicability of the findings to the current situation should be made. Therefore, I suggest the paper should be accepted after some supplementary dissusion was made. 

Reviewer 2 Report

The manuscript entitled “Patterns of perceived indoor environment in Danish homes” try to characterize and identify potential patterns of perceived indoor environments.

 This is in principle an interesting topic, however unfortunately I cannot recommend publication in the journal International Journal of Environmental Research and Public Health:

-          In general, the study only shows results and does not make any contribution of interest, new and/or additional.

-        -  Abstract poorly written. It is redundant, i.e. the last three lines repeat what was said in previous lines. A main summary of the observed results should also be provided.

-     -     Introduction. The state of the art and the references to other works exposed in the introduction are very brief and concise. It has to be completed significantly with studies related to the subject. This will allow the authors to broaden the perspective to approach the article. The last paragraph of the introduction is too short, it is recommended to extend it to properly close the section.

-      -    Materials. “2.2 Assessment of demographic data and health behaviour”. It is a mistake to determine a person's health status solely by considering BMI and smoking status.

-      -    How is the model of the questionnaire? Only some questions and answers are detailed in the text. There is not enough information to assess this point.

-       -   In general, the results presented do not follow a coherent discourse. The presentation of results must be expanded, and an additional contribution must be made in relation to the survey data.

-    -      The format of the figures and their representation is not adequate. i.e Figure 2 and 3 are not understandable (in the text it says 4 factors and it is shown 13 factors, etc.)

Reviewer 3 Report

This paper statistically analyzed and classified the issues related to perceived air quality. My concern is most on the data used in this study, which were achieved from a questionnaire investigation over 20 years ago. There are great changes in the past 20 years in technologies, tobacco restriction, and administrative management. The data on the building construction period, smoking habits, residential density, noise, etc. may have been greatly changed since then. So I have questions about the timeliness of this questionnaire. The author shall clarify if the results are still valid today.